# Establishment and Application of a Quantitative PCR Method for *E248R* Gene of African Swine Fever Virus

**DOI:** 10.3390/vetsci9080417

**Published:** 2022-08-08

**Authors:** Liwei Li, Nannan Du, Jinxia Chen, Kuan Zhang, Wu Tong, Haihong Zheng, Ran Zhao, Guangzhi Tong, Fei Gao

**Affiliations:** 1Shanghai Veterinary Research Institute, Chinese Academy of Agricultural Sciences, Shanghai 200241, China; 2Jiangsu Co-Innovation Center for the Prevention and Control of Important Animal Infectious Disease and Zoonoses, Yangzhou University, Yangzhou 225009, China; 3Xiamen Center for Animal Disease Control and Prevention, Xiamen 361009, China

**Keywords:** African swine fever virus (ASFV), *E248R* gene, real-time PCR, recombinant PRRSV

## Abstract

**Simple Summary:**

African swine fever (ASF) is the most serious animal disease that endangers the pig industry in China, causing the heaviest economic loss so far. Effective prevention and control of ASF is necessary for China’s pig industry, and also the focus and hotspot of global swine industry disease prevention and control research. In view of this, rapid, specific and sensitive diagnosis of ASF is of great significance. In this study, ASF virus (ASFV) *E248R* gene was selected to be the target for establishing a real-time PCR method which did not cross-react with other porcine viruses that could cause similar symptoms. The detection methods can be used for the efficient detection of ASFV infection and the recombinant live-vectored virus-expressing antigen protein of ASFV.

**Abstract:**

ASF has caused huge economic losses to China’s swine industry. As clinical symptoms of ASF were difficult to distinguish from classical swine fever and porcine reproductive and respiratory syndrome (PRRS), rapid and effective differential diagnosis of ASFV seems very important to control the spread of the disease. In this study, the ASFV *E248R* gene was selected to be the target for establishing a real-time PCR method. TaqMan real-time PCR for the detection of ASFV *E248R* gene did not cross-react with other porcine viruses that could cause similar symptoms. The results of the repeatability test showed that the coefficients of variation between and within groups were lower than 1.977%. This method can be used for the rapid detection and early diagnosis of ASF. Meanwhile, the recombinant PRRS virus (PRRSV)-expressing *E248R* gene of ASFV was constructed and rescued by using the reverse genetic platform of live-attenuated PRRSV vaccine. The ASFV *E248R* gene can be detected by using this real-time PCR detection method, confirming that the ASFV *E248R* gene could be stably amplified in PRRSV genome at least 20 cell passages. The detection methods can be used for the efficient detection of the ASFV infection and recombinant PRRSV live vector virus-expressing ASFV antigen protein.

## 1. Introduction

African swine fever (ASF) causes high fever, internal organ bleeding and other clinical symptoms, which can affect pigs of all ages. The clinical manifestations of ASF are various, including acute type, subacute type, chronic type and no obvious clinical symptoms, among which acute type is the most common clinical manifestation, with a mortality rate of 100% [1,2,3,4].

ASF was first discovered in Kenya in 1921 [5]. In 2007, Georgia reported the first case of ASF, which spread to the Caucasus. In the same year, the ASF epidemic was discovered in Russia, and ASF gradually entered Eastern Europe from Russia [6]. In August 2018, the first ASF epidemic occurred in China, which spread to 32 provinces, cities and autonomous regions in only one year, causing huge economic losses to China’s pig industry [7]. Because there is no effective vaccine or drug available for this disease, strict biosafety measures are the most effective way to prevent and control its spread. 

African swine fever virus (ASFV) is the causative agent of ASF, which is a double-stranded DNA virus and the only member of the Asfarviridae family [8]. ASFV virus particle is an icosahedron structure composed of concentric layers with a diameter of about 175–215 nm, with an internal core with a diameter of about 70–100 nm inside, and the outer part is sequentially wrapped by a core shell, an inner envelope, a capsid and an outer envelope [1,9]. The core consists of a nucleoid, including virus genome and some nucleoproteins, such as DNA binding protein p10. ASFV contains a linear double-stranded DNA genome with a length of about 170–194 kb [8]. The two ends of the DNA molecule form a closed hairpin loop through partial base pairing, with a left variable region of about 35 kb, a middle-conserved region of about 125 kb and a right variable region of about 15 kb. The variable regions at both ends contain repetitive sequences [4]. Genome lengths of strains are different due to the addition or deletion of terminal repeats [1]. The genome of ASFV contains 150–167 open reading frames (ORFs), which are involved in the process of virus replication and infection through coding translation into proteins, including about 54 structural proteins and more than 100 non-structural proteins [10,11]. Among them, several studies have shown that ASFV pE248R protein is a complete myristoylation membrane protein, located in the inner membrane of virus, and is a late structural component of virus particles. The amino acid sequence of the pE248R protein contains an N-terminal N myristoylation site (2GGSTSK7) and a C-terminal putative transmembrane region (194SAVFKNIMVAAVVIVLIIVGFIA216) [12]. Recombinant pE248R protein played important role in ASFV replication, manifested in an increase in viral reproduction level [13]. A previous study showed that the deletion of the E248R protein did not affect the virus assembly, but the infectivity of the virus was 100 times lower than that of wild-type virus [14]. It was found that the pE248R protein was a necessary protein in the process of virus–cell fusion and core transfer to cytoplasm [15]. It was shown that the offspring of pE248R deficient virus was non-infectious. pE248R protein plays an important role in the process of virus invading cells, so it has important research value. At the same time, the reduction in viral infectivity by *E248R* gene deletion also provides a new idea for the research of the ASF vaccine and antiviral drugs [12]. In this study, the full-length sequence of ASFV *E248R* gene was inserted between ORF1b and ORF2 of PRRSV vaccine strain HuN4-F112, and the recombinant PRRSV containing ASFV *E248R* was successfully rescued, which would be used in the development of the recombinant live vector vaccine against ASF.

In order to establish a rapid diagnostic method for detecting ASFV and the recombinant PRRSV-expressing ASFV *E248R* gene, specific primers and probes were designed and synthesized according to the conserved sequence region of *E248R* gene of ASFV SY18 strain. A TaqMan real-time fluorescent quantitative PCR detection method of ASFV was established, and its specificity, repeatability and sensitivity were verified. The experimental results shown that this method had good sensitivity, high specificity and no cross reaction with other swine-derived viruses. Repeatability test results show that the repeatability between groups and within groups was good. This method could be used for the rapid diagnosis of ASF and the detection of recombinant PRRSV live vector vaccine-expressing ASF *E248R* gene.

## 2. Materials and Methods

### 2.1. Construct of Plasmid Containing the E248R Gene

The whole gene of *E248R* of ASFV (747 bp) was synthesized artificially by a Personalbio (Shanghai, China) based on the reference sequences available in GenBank (Accession number: MH766894), and then cloned into pUC57 Vector, named pUC57-*E248R*. The artificial pUC57-*E248R* construct was amplified in *E. coli* Top 10, purified with the Plasmid MiniPrep Kit (Qiagen, Hilden, Germany), and quantified using a ND−2000 spectrophotometer (NanoDrop, Wilmington, DE, USA). Then, the copy numbers were calculated using the following formula: Amount (copies/μL) = [DNA concentration (g/μL) × 6.23 × 10^14^]/(plasmid length in base pairs × 324.5 × 2). Ten-fold dilutions of the pUC57-*E248R*, ranging from 10^8^ to 10^2^ copies/μL, were prepared in nuclease-free water and aliquots of each dilution were stored at −40 °C until use.

### 2.2. Virus Strains and Viral Samples

The inactivated antigens of classical swine fever virus (CSFV), porcine reproductive and respiratory syndrome virus (PRRSV), porcine pseudorabies virus (PRV) and porcine epidemic diarrhea virus (PEDV) were kept in our laboratory. Viral DNA and RNA were extracted using the QIAmp Viral DNA/RNA kit (Qiagen, Hilden, Germany) according to the manufacturer’s instructions. Viral RNA was reverse transcribed to cDNA using Primescript II 1st strand cDNA Synthesis kit (TaKaRa, Dalian, China). All of the DNAs and cDNAs templates were stored at −80 °C until use.

### 2.3. Primers and Probe for the Real-Time PCR

The primers and TaqMan probe were designed using Oligo7 (Oligo, Fort Wayne, IN, USA) software to amplify a conserved 126bp fragment within the *E248R* gene. The information of the forward primer E248R-F, the reverse primer E248R-R and the probe are shown in Table 1. The primers were synthesized by Tsingke (Beijing, China). The probe E248R-P was synthesized by GenePharma (Shanghai, China).

### 2.4. Formulation of Standard Curve

Real-time PCR assays were relative fluorescence quantification which were performed on LightCycle96 (Roche, Basel, Switzerland). The 20 μL reaction mixture consisted of 12.5 μL of Premix Ex Taq (Takara, Dalian, China), 1 μL of plasmid DNA or 1 μL of extracted DNA template or 1 μL of cDNA template, 0.6 μL of each primer (10 μmol/L), 0.4 μL of probe (10 μmol/L) and nuclease-free water were put into the reaction system to optimize the assay (Table 2). The PCR conditions included 95 °C for 1 min (initial denaturation), followed by 40 cycles of 95 °C for 5 s (denaturation) and 60 °C for 30 s (annealing). Ten-fold serial dilutions of pUC57-*E248R,* ranging from 10^8^ to 10^2^ copies/μL, were tested to generate the amplification curve. The plasmid copy number logarithm was plotted against the corresponding Cq values and the standard curve was constructed. Nuclease-free water was used as the negative control in the qPCR assay. All of the reactions were conducted in triplicate.

### 2.5. Optimization of Reaction Conditions

According to the designed Tm value of the primer, five different annealing temperatures of 56 °C, 57 °C, 58 °C, 59 °C and 60 °C were selected to perform fluorescent quantitative PCR amplification on the standard of 10^5^ copies/μL. The best annealing temperature was obtained by comparing the Ct values of the same template amplified at different annealing temperatures.

### 2.6. Specificity, Sensitivity and Repeatability Analysis

The specificity of the qPCR was analyzed with the DNA or cDNA of other swine-derived viruses, including CSFV, PRRSV, PEDV and PRV. Nuclease-free water was used as a negative control. The plasmid pUC57-*E248R*, ranging from 10^7^ to 10^1^ copies/μL, was prepared to determine the sensitivity of the reaction. To evaluate the repeatability of the real-time PCR, 10^7^, 10^5^ and 10^3^ copies/μL of pUC57-*E248R* were used to determine the coefficient of variation (CV). For intra-assay variability, triplicates of each dilution were examined, and the CVs were calculated according to the formula: CV = the geometric mean Cq values/standard deviation. For inter-assay variability, the geometric mean of the Cq values and standard deviations from triplicate assays were calculated.

### 2.7. Viral Rescue and Characteristic Analysis of Recombinant PRRSV

The recombinant PRRSV-expressing ASFV *E248R* gene was constructed and rescued on MARC-145 cells by transfected PRRSV full-length cDNA clone inserted the ASFV *E248R* gene. The detailed information could be enquired about in our last study [16]. Cells were monitored daily for cytopathogenic effects (CPE) when the recombinant virus was rescued. The multistep growth curve was conducted in MARC-145 cells as previously described to characterize the viral properties and viral propagation of recombinant virus and parental virus. Expression of PRRSV N and ASFV *E248R* genes in rPRRSV-E248R was determined by indirect immune fluorescent assay (IFA). Cell monolayers at 48 h post-infection were fixed with ice-cold methanol for 10 min at room temperature, blocked with 0.1% bovine serum albumin for 30 min, and incubated with a monoclonal antibody against N (SR30A, Rural Technologies) at 1:5000 or E248R (prepared by our lab, unpublished data) at 1:1000 at 37 °C for 2 h. After washing five times, cells were incubated at 37 °C for 1 h with Alexa Fluor 488-labeled (anti-N) and 568-labeled (anti-E248R) goat anti-mouse IgG (H + L) (Thermo Fisher Scientific, Waltham, MA, USA). After washing, fluorescence was visualized using an inverted fluorescence microscope (model IX71; Olympus, Corporation, Tokyo, Japan), as previously report.

### 2.8. Detection of ASFV Gene in the Samples and Recombinant Viruses

Pig serum samples was collected from pig farms in Zhejiang Province and Fujian Province, China. All experimental programs involving the samples were carried out in accordance with the Guidelines for the Nursing and Use of Experimental Animals and approved by the Ethics Committee of Shanghai Veterinary Research Institute, Chinese Academy of Agricultural Sciences. Viral DNA extracted from 248 sample sera was detected by the developed real-time PCR methods. VetMAX ASFV detection kit was purchased from Thermo Fisher Scientific (Waltham, MA, USA). ASFV ELISA kit was purchased from IDvet (Innovative-diagnostics, Grabels, France). Viral supernatant of rPRRSV-E248R was collected and RNA (Qiagen) extraction kit was used to extract viral RNA according to the instructions. In total, 16 μL viral RNA was added with 4 μL reverse transcription reagent, incubated at 37 °C for 15 min, and the reverse transcriptase was inactivated at 85 °C to complete the reverse transcription process. Using the reverse transcribed virus cDNA as template, the method established in this study was used for foreign gene detection.

## 3. Results

### 3.1. Establishment of Standard Curve

Ultramicro ultraviolet spectrophotometer (ND-2000) was used to measure the concentration of the constructed plasmid pUC57-*E248R*. Because the concentration was 123 ng/μL and the ratio of OD260/280 was 1.846, the plasmid purity met the requirements. The calculated copy number of plasmid is 3.85 × 10^10^ copies/μL. First, the plasmid was diluted to a concentration of 1 × 10^10^ copies/μL, and then diluted by times of ten in turn. Then, the plasmid with a dilution concentration of 1 × 10^8^ to 1 × 10^2^ copies/μL was detected by real-time fluorescence quantitative PCR. The detection results (Figure 1A) showed that the amplification curve of 1 × 10^8^ to 1 × 10^2^ copies/μL positive template plasmid was a typical S. The intervals between curves were even, so the diluted and preserved positive plasmid of 1 × 10^8^ to 1 × 10^2^ copies/μL could be used as the standard of ASFV real-time fluorescence quantitative PCR. The standard curve generated showed a range of 1 × 10^8^ to 1 × 10^2^ copies/μL, with a linear correlation (R^2^) of 1.00, and efficiency of 1.93 between the Cq value and the logarithm of the plasmid copy number. The reaction efficiency of the assay using the slope (the slope was −3.4905) and the Y-intercept was 42.40 from the linear equation. The standard curve was automatically generated by Roche fluorescence quantitative analyzer (Figure 1B). The standard curve equation was y = −3.4905x + 42.40, R² = 1.00.

### 3.2. Optimization of Reaction Conditions

According to the designed Tm value of the primer, five different annealing temperatures of 56 °C, 57 °C, 58 °C, 59 °C and 60 °C were selected to perform fluorescent quantitative PCR amplification on the standard of 10^5^ copies/μL. The best annealing temperature was obtained by comparing the Ct values of the same template amplified at different annealing temperatures. The experimental results showed that when the annealing temperature was 57 °C, the average Ct value of 10^5^ copies/μL standard was 24.67, which was the minimum of the average Ct values among the five annealing temperatures, and the amplification efficiency was the best, so 57 °C was selected as the annealing temperature of fluorescence quantitative PCR. The cycle conditions of fluorescence quantitative PCR reaction were indicated in Table 3.

### 3.3. Analysis of Repeatability, Sensitivity and Specificity

The results of intra-group repeatability test showed (Table 4) that the coefficient of variation in the ASFV real-time fluorescence quantitative PCR was between 0.139 and 1.239%. The results of inter-group repeatability test (Table 5) showed that the coefficient of variation in the fluorescence quantitative PCR was between 1.286 and 1.977%. The intra-group coefficient of variation and inter-group coefficient of variation were lower than 1.977%, and this method had good repeatability. The results of fluorescence quantitative PCR with 1 × 10^0^–1 × 10^8^ copies/μL positive standard plasmid as a template. The minimum detection amount of this method was 1 × 10^1^ copies (Figure 1C), indicating that the fluorescence quantitative detection method of ASFV established in this study had high sensitivity. The nucleic acid samples of CSFV, PRRSV, PEDV, PRV and 10^5^ copies/μL standard were used for simultaneous detection. The results showed that only 10^5^ copies/μL standard could generate specific amplification curves, and other virus samples showed no signal amplification (Figure 1D). The experimental results showed that the probes and primers designed in this study could not identify CSFV, PRRSV, PEDV and PRV virus sequences, and this method did not cross-react with these pathogens and had good specificity.

### 3.4. Detection of the Serum Samples and Recombinant PRRSV-Expressing ASFV E248R

The prepared samples were used for the fluorescence quantitative PCR detection with optimized reaction system and reaction conditions. In total, 248 samples were identified, including 232 negative samples and 16 positive samples. The results were consistent with the results of the commercial quantitative PCR kit from Thermo Fisher Scientific and the commercial ELISA kit from IDvet (Grabels, France). The experimental results showed that the fluorescence quantitative PCR method of ASFV could be applied to the detection of the serum samples.

Recombinant PRRSV-expressing ASFV E248R was rescued on MARC-145 cells and named as rPRRSV-E248R. The obvious CPE could be observed (Figure 2A). IFA for rPRRSV-E248R and its parental virus rHuN4-F112 was performed. The results shown that specific fluorescent signal for ASFV E248R and PRRSV N could be detected in rPRRSV-E248R-infected cells, while for rHuN4-F112-infected cells, only a specific fluorescent signal for PRRSV N could be detected (Figure 2B). Multistep growth curve comparison for rPRRSV-E248R and rHuN4-F112 showed that the viral propagation characteristics was indistinguishable, as shown in Figure 2C. RNA from the viral supernatant was extracted and inverted into cDNA, and the method established in this study was used for detection. The results showed that ASFV *E248R* gene could be detected in the virus cDNA, and the concentration was about 4.64 × 10^5^ copies/μL (Figure 3A). MARC-145 cells were infected by the recombinant virus rPRRSV-E248R, and the supernatants were collected at 2, 12, 24, 48 and 60 h after infection. Then, RNA was extracted, and samples were detected by cDNA after reverse transcription. As shown in Figure 3B, with the progress of virus infection, the virus copy number increased obviously, proving that the fluorescent quantitative PCR detection method can be used to detect the proliferation level of the recombinant virus. By the serial cell passage of recombinant virus in vitro, the genetic stability of the ASFV *E248R* gene (P5, P10, P15, and P20 viral stocks) was detected by the fluorescent quantitative PCR detection method. The results showed that the ASFV *E248R* gene could be stably detected in the recombinant PRRSVs (Figure 3C). IFA for rPRRSV-E248R (P5, P10, P15 and P20 viral stocks) demonstrated the E248R protein could be stably expressed in recombinant PRRSVs (Figure 3D), which was consistent with the results in Figure 3C.

## 4. Discussion

Since the outbreak of ASFV in China in 2018, the etiology research of ASFV has been carried out all over the country [7]. ASFV has huge particles and numerous encoded proteins [8]. At present, most of the research on protein functions remains largely unknown, and some of the existing research studies related to proteins involve relatively little research on the function of E248R protein. Although the research on ASFV has made some progress, there is still no commercial vaccine [17]. It is particularly important to develop effective commercial ASF vaccine and anti-ASFV drugs. It is reported that the incubation period of ASFV infection is 4–19 days. Infected animals spread the virus within two days, show clinical signs and displayed seropositivity on 7–9 days after infection [2]. Therefore, for the early diagnosis of ASF, antigen detection is important. TaqMan fluorescence quantitative RT-PCR, as a specific, sensitive and rapid detection method, is an essential detection method for the early diagnosis of ASF.

The real-time PCR methods developed in recent years have been commonly used due to their high sensitivity, efficiency and specificity. Most of them that have been designed depend on a highly conserved ASFV p72-coding region coding the major viral structure protein [18]. However, there are relatively few detection methods that target other genes. In this study, we established TaqMan-based real-time PCR for detecting ASFV *E248R* gene with high sensitivity, specificity and repeatability. The results showed that the standard curve shared a R^2^ of 1.00 and efficiency of 1.93. This method only specifically amplified ASFV, and did not cross-react with CSFV, PRRSV, PEDV and PRV. The detection limit for ASFV was 10 copies (Figure 1). A previous study showed that a sensitive TaqMan PCR was developed for the detection of the ASFV P72 gene, which is possible to obtain a reliable laboratory diagnosis within 24 h of sample receipt. This method is much faster than virus isolation [19]. Compared with it, the method established in this study is more sensitive. Similar to the real-time fluorescence quantitative PCR method based on *A137R* gene established by Yin et al., the sensitivity test showed that the lowest detection amount was 10 copies, which showed higher sensitivity [18]. Repeatability test results show that the coefficient of variation between groups and within groups is less than 1.977%, and the repeatability is good (Table 5). Through the detection of the serum samples, it can be seen that the real-time fluorescence quantitative detection method established in this study can effectively detect positive nucleic acid samples, and the results are consistent with the commercial quantitative PCR and ELISA detection kits. Sample extraction and fluorescence quantitative PCR detection can be completed in 2 h at the earliest. Therefore, the TaqMan real-time fluorescence quantitative detection method based on ASFV *E248R* gene established in this study was a specific, sensitive and efficient TaqMan real-time fluorescence quantitative PCR detection method that can be applied to clinic detection.

Studies have shown that the deletion of the E248R protein can effectively reduce the infectivity of ASFV, which also shows that the E248R protein is a promising target for developing gene-deleted ASF strains or anti-ASFV drugs [10,12,13]. The preparation of ASFV protein and antibody can provide a basis for the development of diagnostic reagents and subunit vaccines. Up to date, the research on the related functional mechanism of E248R protein is not enough to support the future research and development. The polyclonal antibody of E248R prepared in this experiment is helpful to further study the related functional mechanism of protein in ASFV and lays a foundation for the research and development of diagnostic reagents related to ASFV.

Since the E248R gene is considered to be a good antigen gene, we constructed the recombinant PRRSV-expressing E248R gene (Figure 2). The recombinant virus is an important portion of the live-vectored vaccine against ASF developed in our lab. So far, there is no fluorescence quantitative PCR method for this gene. In this study, we developed a method for this gene with excellent specificity and sensitivity. At the same time, this method is also used to detect the recombinant PRRSV-expressing E248R gene, which can detect the nucleic acid copy number of the E248R gene in recombinant viruses (Figure 3).

The TaqMan real-time fluorescent quantitative PCR detection method developed in this study was determined to detect ASFV. This method had good sensitivity, high specificity, good repeatability and no cross reaction with other swine-derived viruses, which provided a practical, simple, economical and reliable tool for the rapid diagnosis of ASF and the detection of a recombinant PRRSV live vector vaccine-expressing ASF *E248R* gene.

## Figures and Tables

**Figure 1 vetsci-09-00417-f001:**
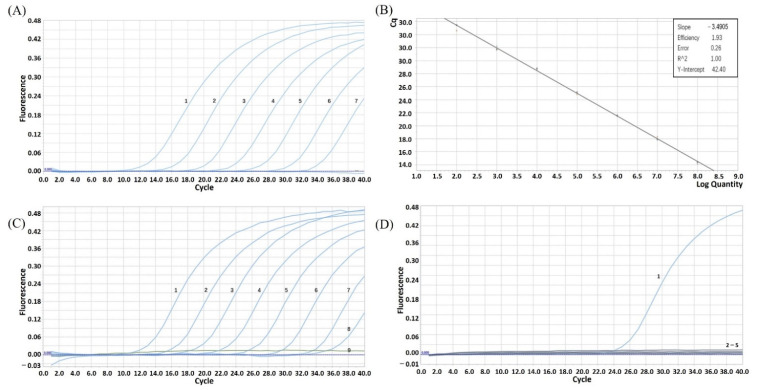
(**A**) Amplification curve of ASFV *E248R*-positive standard plasmid. 1–7. 1 × 10^8^~1 × 10^2^ copies/μL plasmid template. (**B**) The standard curve of ASFV *E248R* gene TaqMan real-time PCR. (**C**) ASFV *E248R* gene TaqMan real-time PCR amplification curve of sensitivity test. 1–9. 1 × 10^8^~1 × 10^0^ copies/μL plasmid template. (**D**) The specificity test of ASFV *E248R* gene TaqMan real-time PCR. 1. Positive plasmid; 2–5. Nucleic acid samples of CSFV, PRRSV, PEDV and PRV.

**Figure 2 vetsci-09-00417-f002:**
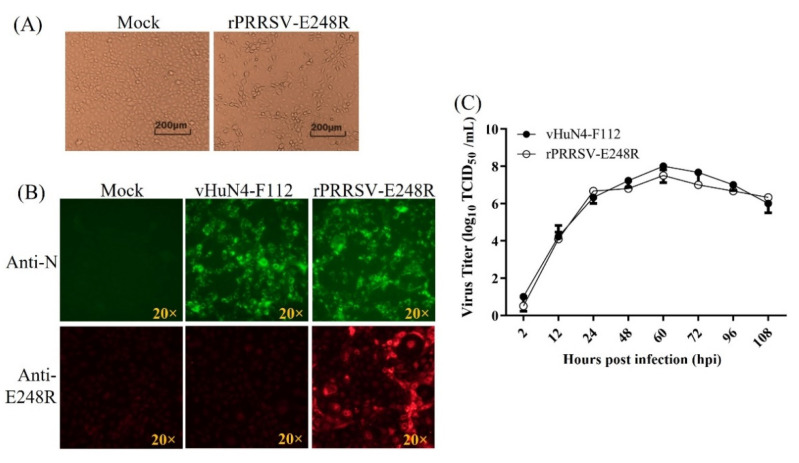
(**A**) Viral rescue for rPRRSV-E248R. CPE observation for Mock and rPRRSV-E248R-infected MARC-145 cells. (**B**) Immunofluorescent assay (IFA) for identification of PRRSV N and ASFV E248R. At 50 h post-infection (hpi), MARC-145 cell monolayers were fixed and stained with N protein-specific and ASFV E248R-specific antibodies using Alexa Fluor 488 goat anti-mouse IgG (H + L) and 568 goat anti-mouse IgG (H + L) secondary antibodies, respectively. Specific fluorescent signal was viewed with an immunofluorescence microscopy. The IFA patterns of parental (rHuN4-F112) and recombinant viruses (rPRRSV-E248R) were shown. (**C**) Multistep growth curves. A multiplicity of infection (MOI) of 0.01 for the rHuN4-F112 and rPRRSV-E248R was used to infect fresh MARC-145 cells. Supernatants were harvested at 2, 12, 24, 48, 60, 72, 96 and 108 hpi. Viral titers were determined by the TCID_50_ and expressed as log_10_TCID_50_/mL.

**Figure 3 vetsci-09-00417-f003:**
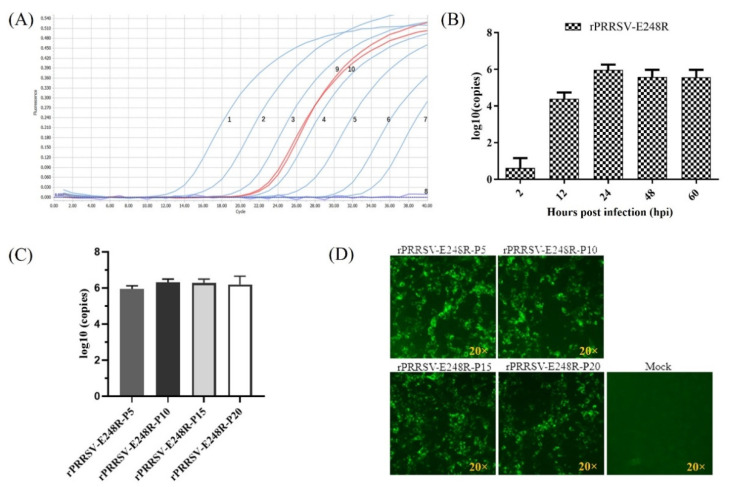
(**A**) Results of viral propagation by real-time PCR. 1–7. 1 × 10^8^–1 × 10^2^ copies/μL plasmid template: 8. Negative control: 9–10. Viral cDNA. (**B**) Viral replication tendency detection by the fluorescent quantitative PCR detection method. The recombinant virus rPRRSV-E248R-infected MARC-145 cells. Cell culture supernatants were collected at 2, 12, 24, 48 and 60 h after infection. Then, RNA was extracted, and samples were prepared and detected by cDNA after reverse transcription. (**C**) The copy level of *E248R* gene was detected in the recombinant PRRSVs (P5, P10, P15, and P20 viral stocks) by the fluorescent quantitative PCR detection method. (**D**) The E248R protein expression in the recombinant PRRSVs (P5, P10, P15 and P20 viral stocks) was detected by IFA.

**Table 1 vetsci-09-00417-t001:** Sequence of primer pairs and probe which were used in the study.

Name	Primers 5′-3′	Position
E248R-F	5′-GGAGGCTCTACAAGCAAA-3′	166996–167013
E248R-R	5′-CATCACCGAATACGCCTA-3′	167105–167122
E248R-P	5′FAM-AATACGACCAACATTATCAGCAAT-3′BHQ1-3′	167025–167050

**Table 2 vetsci-09-00417-t002:** Probe qPCR Mix (2×) enzyme reaction system of 20 μL.

System Components	Volume
Probe qPCR Mix (2×)	10 μL
probe	0.4 μL
Primer-F	0.6 μL
Primer-R	0.6 μL
Template	1 μL
Nuclease-free water	7.4 μL
Total	20 μL

**Table 3 vetsci-09-00417-t003:** Ct values of amplification curve of different annealing temperatures.

Tm (°C)	Ct	Average Value ± Standard Deviation
56	24.95	24.927 ± 0.068
	24.98	
	24.85	
57	24.70	24.670 ± 0.030
	24.67	
	24.64	
58	24.47	24.767 ± 0.270
	25.00	
	24.83	
59	24.83	25.097 ± 0.326
	25.00	
	25.46	
60	24.22	25.25 ± 0.198
	25.11	
	25.39	

**Table 4 vetsci-09-00417-t004:** Intra-assay repeatability of ASFV *E248R* gene TaqMan real-time PCR.

Standard Plasmid	Ct	Average Value	Standard Deviation	CV%
10^5^	24.82	25.017	0.174	0.695
	25.15			
	25.08			
10^6^	21.49	21.520	0.030	0.139
	21.55			
	21.52			
10^7^	17.83	17.997	0.223	1.239
	17.91			
	18.25			

**Table 5 vetsci-09-00417-t005:** Intergroup repeatability of ASFV *E248R* gene TaqMan real-time PCR.

Standard Plasmid	Ct	Average Value	Standard Deviation	CV%
10^5^	24.82	24.303	0.480	1.977
	24.22			
	23.87			
10^6^	21.49	21.700	0.279	1.286
	21.86			
	21.79			
10^7^	17.83	17.540	0.252	1.439
	17.42			
	17.37			

## Data Availability

The data that support the findings of this study are available from the corresponding author upon reasonable request.

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
