# Peer review of "Establishment and Application of a Quantitative PCR Method for *E248R* Gene of African Swine Fever Virus"

_vetsci, 2022, doi:10.3390/vetsci9080417_

Round 1

Reviewer 1 Report

The study authored by Liwei et al., investigated the development of diagnostic approach on E248R gene of ASFV by the application of fluorescence quantitative PCR. The methodology and result part was well explained, but the discussion part needs few more points. However, few correction and clarifications are necessary to improve the article.

Major comments

The authors need to add few more points about the structural and functional importance of E248R and the myristoylation site, C-terminal site (sequence) info from the citation 12 (Rodriguez et al., 2009).

The authors should verify the primer positions, and confirm the primer position from E248R. since the NCBI-Genebank showed E248R sequence starts at 166927-167772. However, the primer position varies in the table 1.

Figure 1. The figures are not clear, X and Y axis numbers and label are not visible. The authors should present the figures like the previous published article (Yin et al., 2019) regarding A137R gene on ASFV.

The authors must discuss the present result with the previous findings and to explain why the E248R protein expression in the recombinant PRRSV results important and discovering what?

Minor comments

Simple summary: remove "endangers" word and change to "impact on pig industry" or modify "endangers the pig industry"

Introduction: Line 61: modify to "Genome lengths of strains are different due to"

Methods: Line 94: Provide additional information of DNA/protein fasta sequence of E248R in the supplementary file to justify the line 82 (the conservation profile of sequence).

Line 119: Provide model, company info of real time PCR

Line 153: Typo error: "37 C," write correctly.

Line 179 and 180: Write correctly and follow same as line 175 (1x108 to 1x102 copies/ul)

Table 6: Already explained in the text line 225. The table has not much information else. Remove or provide in the supplementary file.

Reviewer 2 Report

The study is of scientific interest since great ASF virus has been detected in swine in many countries. In general the manuscript is clear, but there are some parts that can be improved, including the description of the origin of the samples and biosecurity managment of the samples in the Lab.

Title, “…Establishment and application of a new fluorescence quantitative PCR method for E248R gene of African swine fever virus” should be delete "new fluoresence"  maybe more suitable for this manuscript.

The study describes E248R gene as candidate antigens for qPCR. What other gene candidate would be used as diagostic in ASF? What are the adventajes of using it?

The study does not describe any actual diagnostic for ASF. There are numerous candidate genes for diagnostic- whether they actually provide control requires a lot more study than is provided here.

Reviewer 3 Report

The development of a new real-time PCR for E248R gene of ASFV can be useful. 

I did not indicate all English language and style changes required. I suggest you ask for assistance with writing the paper grammatically correct.

Simple summary, line 18-19: Either you detect ASFV or not. I would not say "clinical" as this would mean you suspect symptoms of ASF to be present and not all ASFV will give visible clinical symptoms.    

Abstract, line 26-27: Incomplete sentence. It does not state what was shown.

Abstract, line 33: Delete "clinical"

Introduction, line 38: Change "caused high fever" to "causes high fever"

Introduction, line 51: ASF is the only member of the Asfarviridae family there is no "African classical swine fever family".

Materials and methods, section 2.1 and 2.4: In section 2.1 you prepared "107 to 100" in section 2.4 you use "108 to 102" where did you get the extra dilution from? You do discuss this in results line 173-175.

Results, line 171 - 175: Delete "According to the....molecular weight", this is discussed in methodology and does not need to be repeated.

Results, section 3.2: This was not discussed in methodology section. Please include.

Results, line 223: You did not describe the sample collection in methodology section. Please include ethical approval for pig serum collection.

Round 2

Reviewer 1 Report

I appreciate the authors for the efforts to all the correction and provided the response to my questions and concerns.